# Interaction of Copper(II) Ions with Certain Oxyacids and Azoles

Nina Skorik, Evgeniya Tomilova, Ekaterina Pastushchak and Irina Kurzina *

Faculty of Chemistry, National Research Tomsk State University, 634050 Tomsk, Russia
* Correspondence: kurzina99@mail.ru; Tel.: +7-913-882-1028

**Abstract:** Compounds composed of $[Cu(HIm, metIm)_{2-3}L] \cdot nH_2O$ ($n$ = 0, 2) were obtained during the interaction of slightly soluble tartrate and copper(II) salicylate composed of $CuL \cdot nH_2O$ ($n$ = 1–2, $L^{2-}$—$Tar^{2-}$, $Sal^{2-}$) with imidazole (HIm) and 2-methylimidazole (metIm). Mono- and bi-ligand salts were analyzed; the process of their thermal decomposition was studied. The solubility constants $K_S$ of the $CuC_4H_4O_6 \cdot 2H_2O$ tartrate and $CuC_7H_4O_3 \cdot H_2O$ salicylate of copper(II) with at ionic strength of 0.3 were determined. The IR spectroscopy method showed the participation in complexation of the nitrogen atom N(3) of imidazole and the oxygen atoms of the carboxyl groups of oxyacids, as well as the hydroxyl group of salicylic acid in the mixed-ligand salts of copper(II). The compositions and stability of the imidazole-tartrate(salicylate) copper(II) complexes in an aqueous solution were determined by performing photometry and spectrophotometry and of the monoligand complexes [CuTar] and [CuSal] were determined by solubility and isomolar series methods.

**Keywords:** tartrate and salicylate of copper(II); imidazole and 2-methylimidazole; mixed-ligand complexes; stability; solubility

## 1. Introduction

Tartaric $C_4H_6O_6$ ($H_2Tar$), salicylic $C_7H_6O_3$ ($H_2Sal$) acids, and some of their salts with cations of biometals (such as Fe, Zn, Cu, Mn, and Co) are used for the development of new drugs (e.g., the acidic salicylate of copper(II) $Cu(C_6H_4(OH)COO)_2$ is used in dermatology), biologically active substances, and fungicidal compositions [1]. They can be interesting for researchers working on the synthesis of new materials with predictable properties [2].

References [3–5] described the methods of extracting copper(II) salts with tartaric acid of different compositions from aqueous solutions, for example, salts $CuC_4H_4O_6 \cdot 3H_2O$ and $CuC_4H_4O_6 \cdot 2H_2O$ [3]; MTart compounds (where $M^{2+}$ is the ion of Mn, Fe, Co, Ni, Cu, and Zn) [4]; and the coordinated polymer $\{[Cu_2(Tart)_2(H_2O)_2] \cdot 4H_2O\}_n$ [5].

Both phenolic and carboxylic groups are present in salicylic acid, so salicylic acid can form acidic and medium salts. When the copper ion is coordinated by one or two functional groups (−COOH and −OH), homo- and hetero-nuclear compounds are formed. For example, there is the low-soluble copper salicylate $CuC_7H_4O_3 \cdot H_2O$ [1]; compounds $M(HSal)_2 \cdot nH_2O$ ($n$ = 3 for cobalt and copper salts, and $n$ = 2 for the lead salt) [6]; and heteronuclear mixed-ligand salicylates $[CuSr(Ba)(HSal)_4(DMAA)_4H_2O]$ and $[CuCu(HSal)_4(H_2O)_2] \cdot 2DMAA$ (DMAA is dimethylacetamide) [7], which are used as precursors for the synthesis of new compounds.

The complexing of copper(II) with tartaric and salicylic acids in an aqueous solution is investigated. The formation of complexes between ions $Cu^{2+}$ and $Tart^{2-}$ ($L^{2-}$) at 25 °C in a 1 M solution of $NaClO_4$ has been studied [8]. The presence of complexes with different compositions in the pH range of 1–4, namely, CuL, CuHL, $CuL_2$, $CuHL_2$, $CuH_2L_2$, $Cu_2L_2$, $Cu_2L_3$, and $Cu_2L_4$ (no charges listed), has been shown. The logarithms of the stability constants of these complexes have been determined. The data on the stability constants of the tartrate and salicylate copper(II) complexes are presented in [8–10].

In biological systems, metal ions typically interact with several ligands. The anions of carboxylic acids, oxyacids, amino acids, vitamins, and azoles containing donor atoms of

oxygen and nitrogen can participate as ligands in such mixed-ligand metal compounds. Azoles are used to produce anti-infective drugs.

There is information on the syntheses of copper(II) biligand salts, including imidazole(HIm) and its derivatives: $Cu(1,2\text{-dimethylIm})_2(HSal)_2$ and $Cu(2\text{-methylIm})_3Sal$ (1,2-dimethylIm, 2-methylIm, 1,2-dimethylimidazole, and 2-methylimidazole) [11]; $Cu(HIm)_n(Hsal)_2$, where $n$ = 2, 5, 6 was obtained owing to the reaction of imidazole with the $Cu_2(HSal)_4$ salt [12]; hexaki**s**-(N-methylimidazole) copper(II) salicylate and $[Cu(N\text{-methylIm})_6](Hsal)_2$ [13]; and [CuLHSal] (L is deprotonated 4-phenyltyosemycarbazide) [14]. $[Cu(HIm)_2(cinn)_2(H_2O)]$, $[Cu(HIm)_2(paba)_2]$, and $[Cu(HIm)_2(clba)_2]$ have the $cinn^-$ — $C_9H_7O_2{}^-$, $paba^-$ — $C_7H_6NO_2{}^-$, and $clba^-$ — $C_7H_4ClO_2{}^-$ — anions of cinnamon, para-aminobenzoic, and 2-chlorobenzoic acids, respectively [15]. $[Cu(HIm)_6]Cl_2 \cdot 4H_2O$ and $[Cu(HIm)_6]Cl_2 \cdot 2H_2O$ have been synthesized using the hydrothermal method [16].

Both the production of solid mixed-ligand salts (MLSs) and the simultaneous study of mixed-ligand complexes (MLCs) in a solution based on nitrogen and oxygen ligands are of particular interest [17–23]. Therefore, the purpose of this study was to develop a method for synthesizing the mixed-ligand compounds of copper(II) with tartaric and salicylic acid anions and azoles using previously synthesized slightly soluble tartrate and salicylate of copper(II). Another purpose of the research was to establish the physicochemical properties and structure (the mode of coordinating organic ligands using a complexant) of the synthesized compounds, as well as to determine the composition and stability of the biligand imidazole-tartrate(salicylate) complexes $[Cu(HIm)_XL]$ formed in the solution.

## 2. Materials and Methods

The objects of the research are copper(II) compounds with biologically active oxy-carboxylate ligands—tartaric and salicylic acids—and those with azoles—imidazole and 2-methylimidazole—in the form of solid salts and complex compounds in a solution. Imidazole $C_3H_4N_2$ has a five-membered cycle with two heteroatoms of nitrogen and has amphoteric properties.

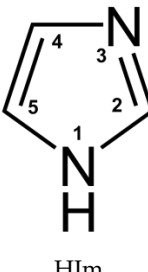

HIm

The nitrogen $N_{(3)}$ atom with an unshared electron pair is capable of protonation ($\lg B_1$ has values in the range of 7–7.7; $B_1$ is the constant of imidazole protonization) and the coordination of metal ions. The compound 2-methylimidazole $C_4H_6N_2$ forms less stable complexes than imidazole because of emerging steric hindrances owing to a substituent [24].

metIm

To analyze and study the properties of the synthesized salts to determine the composition and stability of the mixed-ligand complexes in the solution, the following were used: thermal, thermogravimetric, and elemental analyses; pH potentiometry, photometry,

and spectrophotometry; and IR spectroscopy. The spectra of the solutions were recorded using the spectrophotometer LEKI SS2107UV. The optical density of the solutions was also measured using the photocolorimeter KFK–2–UHL 4.2, with an absorbing layer thickness of $l$ = 10 mm. The pH of the solutions was measured using the pH-meter-673. Its glass electrode was calibrated using buffer solutions with a pH from 3.56 to 6.86. Thermograms of the synthesized salts were recorded using the Netzsch STA 449 F1 device under the following conditions: the crucible material was $Al_2O_3$, the heating rate was 10 °C per min, and the atmosphere was air (80 mL/min). The IR spectra of the ligands and mono- and bi-ligand salts in tablets from KBr were obtained on the spectrometer Agilent Cary 630 FTIR at a frequency of 400–4000 $cm^{-1}$. The synthesized salts were analyzed by means of an automatic elemental CHNS analysis on the analyzer EURO EA 3000 using the microbalance Sartorius MSE 3.6P-000-DM. The used reagents were labeled as ch.p. or p.a. and were not additionally recrystallized.

## 2.1. Synthesis and Solubility of Tartrate and Salicylate Copper(II)

Original monoligand salts of copper(II) were synthesized using aqueous solutions via the reaction between chloride or the nitrate of metal ($CuX_2$) and tartaric or salicylic acid ($H_2L$), partially neutralized by sodium hydroxide, with a mole ratio of $CuX_2:H_2L$ = 1:1

$$CuX_2 + H_2L + (1.8\text{-}1.9)NaOH \rightarrow CuL\downarrow.$$

The precipitates were released at a pH of 4–5 of the mixture. After being rinsed with water, they were dried in the air. By heating the salts at 125 and 900 °C, the content of water and the oxide of copper(II), respectively, was determined. In addition, the metal ion content was found trilonometrically (it satisfactorily coincided with the results of the metal content in oxide). The elemental analysis confirmed the composition of the monoligand salts.

Table 1 shows the data of the analyses of the original salts and the tartrate and salicylate of copper(II). The solubility of the monoligand salts in the (H, Na)$NO_3$ solutions with an ionic strength of $I$ = 0.3 at 25 °C was studied.

**Table 1.** Analysis data of mono- and bi-ligand salts of copper(II) with tartaric and salicylic acids, imidazole, and 2-methylimidazole.

| Compound | N, % | | C, % | | H, % | | CuO, % | | $H_2O$, % | |
|---|---|---|---|---|---|---|---|---|---|---|
| | $f^*$ | $c^*$ | $f$ | $c$ | $f$ | $c$ | $f$ | $c$ | $f$ | $c$ |
| $CuC_4H_4O_6 \cdot 2H_2O$ | – | – | 19.1 | 19.38 | 2.7 | 3.23 | 33.1 | 32.12 | 14.6 | 14.54 |
| $Cu(C_3H_4N_2)_3C_4H_4O_6 \cdot 2H_2O$ | 17.9 | 18.60 | 34.1 | 34.52 | 4.0 | 4.43 | 16.6 | 17.61 | 8.1 | 7.97 |
| $CuC_7H_4O_3 \cdot H_2O$ | – | – | 37.8 | 38.59 | 2.7 | 2.76 | 35.9 | 36.54 | 8.3 | 8.27 |
| $Cu(C_3H_4N_2)_3C_7H_4O_3$ | 21.0 | 20.80 | 46.9 | 47.54 | 3.9 | 3.96 | 21.0 | 19.69 | – | – |
| $Cu(C_4H_6N_2)_2C_7H_4O_3$ | 17.9 | 15.40 | 49.5 | 49.50 | 4.6 | 4.40 | 20.4 | 21.87 | – | – |

$f^*$—found; $c^*$—calculated.

Table 2 shows the data on solubility, the calculation of the solubility constants $K_S$ of the tartrate $CuC_4H_4O_6 \cdot 2H_2O$ and salicylate $CuC_7H_4O_3 \cdot H_2O$ of copper, and the calculation of the stability constants of the complexes of the [CuL] composition, taking into account the hydrolysis of ion $Cu^{2+}$ and the protonization of acid anions. To simultaneously calculate according to the solubility data the solubility constants $K_S$ of the salts composed of 1:1 $CuL \cdot nH_2O$ and the stability constants $\beta_1$ of the complexes composed of 1:1 CuL, formed in the saturated solution, the author program "Solubility" [25] was used. In this program, the following notations were used: $f$—ligand protonization function, $f = 1 + B_1h + B_2h^2$; $B_i$—common constants of anion protonization of oxyacids: hydrolysis function $\omega = 1 + K_{h1}/h$; $K_{h1}$—**hydrolysis constant of $Cu^{2+}$ ion in the first step**, $h$ = [$H^+$]. When using the material balance equations for the metal and ligand (in the saturated solution composed $C_M = C_L$)

and the expression for the constant of heterogeneous equilibrium $K_S$ = [Cu][L] (charges are omitted), the equation for $K_S$, reduced to linearity, can be written as follows:

$$C_{Cu} = [CuL] + \sqrt{K_S} \cdot \sqrt{f}\omega,$$

and the stability constant value of the complex is $\beta_1$ = [CuL]/$K_S$. According to the data in Table 2, the values of $K_S$ and $\beta_1$ were calculated by using the method of minimizing the averaging variance lg$K_{Si}$.

**Table 2.** Data on solubility and calculation of solubility constants $K_S$ of salts $CuC_4H_4O_6 \cdot 2H_2O$ and $CuC_7H_4O_3 \cdot H_2O$ in 0.3 mole/L of solutions (H, Na)NO$_3$ ($K_{h1}$ (Cu$^{2+}$) = 3.1 $\cdot$ 10$^{-8}$; for H$_2$Tar: log$B_1$ = 3.95, log$B_2$ = 6.76; for H$_2$Sal: log$B_1$ = 13.7, log$B_2$ = 16.53).

| № | pH | $C_{\textbf{CuTar}}$, mole/L | $-\log K_S$ (CuTar $\cdot$ 2H$_2$O) | pH | $C_{\textbf{CuSal}}$, mole/L | $-\log K_S$ (CuSal $\cdot$ H$_2$O) |
|---|---|---|---|---|---|---|
| 1 | 4.62 | 3.82 $\cdot$ 10$^{-4}$ | 7.44 | 5.43 | 7.20 $\cdot$ 10$^{-3}$ | 13.57 |
| 2 | 3.50 | 5.03 $\cdot$ 10$^{-4}$ | 7.52 | 5.05 | 7.10 $\cdot$ 10$^{-3}$ | 13.68 |
| 3 | 3.48 | 5.63 $\cdot$ 10$^{-4}$ | 7.46 | 5.14 | 7.53 $\cdot$ 10$^{-3}$ | 13.61 |
| 4 | 3.46 | 6.54 $\cdot$ 10$^{-4}$ | 7.37 | 4.93 | 7.80 $\cdot$ 10$^{-3}$ | 13.65 |
| 5 | 3.14 | 7.85 $\cdot$ 10$^{-4}$ | 7.44 | 4.96 | 8.90 $\cdot$ 10$^{-3}$ | 13.57 |
| 6 | 3.11 | 8.45 $\cdot$ 10$^{-4}$ | 7.41 | 4.48 | 1.15 $\cdot$ 10$^{-2}$ | 13.58 |
| 7 | 2.97 | 9.66 $\cdot$ 10$^{-4}$ | 7.43 | 4.25 | 1.32 $\cdot$ 10$^{-2}$ | 13.60 |

Calculation results: for $CuC_4H_4O_6 \cdot 2H_2O$ log$\beta_1$ (optim.) = 3.74; $\beta_1$ = 4.67 $\cdot$ 10$^3$; log$\overline{K}s$ = −7.44; standard deviation is $s^2$ = 2.68 $\cdot$ 10$^{-3}$; for $CuC_7H_4O_3 \cdot H_2O$ log$\beta_1$ (optim.) = 11.27; $\beta_1$ = 1.86 $\cdot$ 10$^{11}$; log$\overline{K}s$ = −13.61; $s^2$ = 1.77 $\cdot$ 10$^{-3}$.

In the saturated solutions of copper(II) tartrate, the copper concentration was determined iodometrically. The concentration of the solution of sodium thiosulfate was specified in the presence of tartaric acid, which exerts little influence on the results of the iodometrical determination of copper(II). In the saturated solutions of copper(II) salicylate, the copper concentration was determined by using the calibration characteristic obtained via the dependence of the optical density of the solutions on the concentration of the synthesized salt CuSal $\cdot$ H$_2$O (Table 3).

**Table 3.** Composition of solutions for building the calibrating characteristic of system CuSal–NaNO$_3$ ($\lambda$ = 750 nm; pH 3.6; $I$ = 0.3; $C^0$ (CuSal $\cdot$ H$_2$O) = 8.84 $\cdot$ 10$^{-3}$ mole/L; $V_{cum}$ = 6 mL).

| № | $V_{\textbf{CuSal}}$, mL | $C_{\textbf{CuSal}}$, mole/L | $D_{750}$ | Parameters of Straight Line |
|---|---|---|---|---|
| 1 | 1 | 1.59 $\cdot$ 10$^{-3}$ | 0.043 | |
| 2 | 2 | 3.19 $\cdot$ 10$^{-3}$ | 0.060 | |
| 3 | 3 | 4.78 $\cdot$ 10$^{-3}$ | 0.073 | $a$ = 0.0229, |
| 4 | 4 | 6.38 $\cdot$ 10$^{-3}$ | 0.091 | $b$ = 11.222, |
| 5 | 5 | 7.89 $\cdot$ 10$^{-3}$ | 0.112 | $R^2$ = 0.993 |
| 6 | 6 | 9.57 $\cdot$ 10$^{-3}$ | 0.133 | |

### 2.2. Synthesis and Thermogravimetry of Mixed-Ligand Salts of Copper(II)

Mixed-ligand compounds were obtained from the synthesized oxycarboxylates of copper(II) CuL $\cdot$ $n$H$_2$O and imidazole and 2-methylimidazole according to the reaction

$$CuL + xHIm(metIm) \rightarrow Cu(HIm)_X(metIm)_X L\downarrow.$$

The reaction was carried out in an aqueous solution with a pH of 6.5–8.5, and, subsequently, the precipitation of the mixed-ligand salts from the aqueous solution for the

imidazole(methylimidazole)-salicylate salt and from the aqueous–etheric solution for the imidazole tartaric salt was carried out. For this purpose, a weighed portion of the original mono-ligand salt was placed into a small volume of water (8–10 mL). Then, the calculated amount of HIm(metIm) was gradually added to the suspension, creating a molar ratio of CuL:HIm(metIm) equal to 1:1, 1:2, etc. This was carried out until the copper(II) tartrate dissolved completely or the salt of salicylic acid transformed into a new homogeneous, brightly colored bilig and salt. The suspension CuSal–Him(metIm) was held at room temperature for one day. For the mixed-ligand salts, the following procedure was conducted: an elemental analysis; a thermal analysis of the content of water and metal oxide; and a thermogravimetric analysis quantitatively confirming the content of water, ligands, and metal oxide in the salts. Table 4 shows the results of the analysis of the thermograms of the two salts as an example.

**Table 4.** Analysis of thermograms of copper(II) biligand salts.

| № | Nature of Effect | Temperature Interval, °C | Loss of Mass (from init.), % | | Corresponding Process |
|---|---|---|---|---|---|
| | | | *f* | *c* | |
| | | $Cu(HIm)_3Sal$ | | | |
| 1 | Endo-effects | 175–340 | 49.5 | 50.57 | Loss of imidazole ligand |
| 2 | Exo-effects | 340–510; 510–900 | 29.8; 21.0 | 33.70; 19.69 | Destruction of salicylate ion and oxide formation |
| | | $Cu(metIm)_2Sal$ | | | |
| 1 | Endo-effects | 170–340 | 43.3 | 45.09 | Loss of methylimidazole |
| 2 | Exo-effects | 340–510; 510–900 | 33.6; 23.3 | 37.43; 21.87 | Destruction of salicylate ion and oxide formation |

The formulas of the synthesized biligand salts were calculated using the results of the analyses (Table 1). The data on the mass content of the copper(II) oxide and water in the salts were averaged using the results of the thermal and thermogravimetric analyses. Since the removal of water, the degradation of the oxyacid anion, azole, and the formation of the corresponding copper(II) oxide occur in different temperature ranges in the synthesized salts, the data of the thermogravimetric analysis of the biligand salts confirm the salt compositions established using other methods and allow us to suggest the mechanism of their thermal decomposition (Table 4).

So, for example, the thermal decomposition of the imidazolesalicylate of copper(II) $Cu(HIm)_3Sal$ in the air proceeds in several stages, which are quantitatively confirmed by the change in the salt mass (*f*—found; *c*—calculated). The endothermic process of imidazole loss proceeded at 175–340 °C. Moreover, immediately, with the exothermal effect within 340–900 °C, the complete combustion of the salicylate ion and the formation of metal oxide in the air took place. The methylimidazolesalicylate of copper(II) decomposed similarly to the thermal decomposition of the imidazolesalicylate of copper(II). Thermogravimetric studies of synthesized compounds are important for understanding their thermal stability, which, along with other properties, is used by scientists searching for new materials.

### 2.3. Determination of Composition and Stability of Mono- and Bi-ligand Complexes in Solution

The stability constants of the CuL complexes ($L^{2-}$—anion of oxyacid), required for the calculation of the stability constants of MLC, are taken from the literature or calculated based on our data of solubility methods and isomolar series. The composition of the monoligand CuL complexes, formed in the solution, was determined by using the method of isomolar series. According to the data of the isomolar series of the $CuCl_2$—NaHSal

system (Figure 1), whose diffused maximum points to the presence of two complexes in the solution, the stability constant of the [CuSal] complex was calculated.

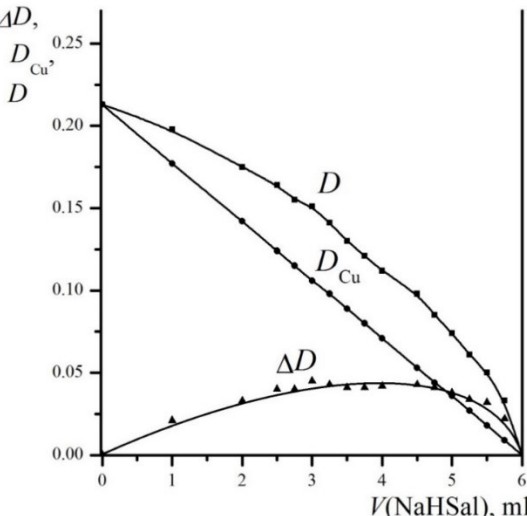

**Figure 1.** Isomolar series of system CuCl₂—NaHSal: ($C^0$ (CuCl₂) = $C^0$ (NaHSal) = $2 \cdot 10^{-2}$ mole/L, $V_{\text{cum}}$ = 6 mL, pH 3.5, $I$ = 0.3, $\lambda_{\text{ef}}$ = 750 nm, $l$ = 10 mm).

To calculate the stability constant of the [CuSal] complex according to the isomolar series data, the experimental data with a ratio of $C_L$:$C_M$, being in the 0.5–1.9 interval, were used. The [CuSal]$^0$ complex dominance in this range was confirmed by the permanence of the stability constant of the complex (Table 5). According to the isomolar series data (Table 5), the stability constant of the [CuSal] complex was calculated by using the following formula:

$$\beta_1 = ([CuSal])/([Cu^{2+}] \times [Sal^{2-}]) = (C_C \omega f)/\{(C_{Cu} - C_C) \times (C_{Sal} - C_C)\},$$

where $C_C$, $\omega$, and $f$ are the complex concentration in the equilibrium solution, the Cu²⁺ cation hydrolysis functions, and the protonization of the salicylic acid anion, respectively: $C_C = \{(D - D_{Cu})C_{Cu}\}/(D_\infty - D_{Cu})$ when $C_{Cu} < C_{Sal}$; $[Sal^{2-}] = (C_{Sal} - C_C)/f$, $[Cu^{2+}] = (C_{Cu} - C_C)/\omega$; $\alpha_C = C_C/C_M$ ($C_L$) complex yield. $D_\infty$ was assessed in the solutions with a molar ratio of Cu²⁺:L²⁻ = 1:1 and a pH value varying within 3–6.

According to the data of the saturation curves of the Cu²⁺–L²⁻–HIm systems (Figures 2 and 3), the stability constants $\beta_{111}$ and $\beta_{121}$ of the mixed-ligand complexes of the composition [Cu(HIm)₁,₂L] were calculated. The saturation curve (Figure 2) is wavy, having a plateau, which is evidence of the stepwise nature of complexing. The presence of the plateau in Figure 2 (the ratio of $C_{HIm}$:$C_{Cu}$ is in the interval of ≈(0.7–1.2)) indicates that, in this area, the complex composition 1:1:1 ([CuHImTar]) dominates, which is confirmed by the constancy of the value of log$\beta_{111}$ (Table 6).

The absence of points in Figure 3 up to the $C_{HIm}/C_{Cu}$ ratio ≈1.7 is associated with the precipitation of low-soluble copper(II) salicylate, which is then dissolved by increasing the imidazole concentration. The stability constant $\beta_{121}$ of the complex with a composition of 1:2:1 ([Cu(HIm)₂Sal]) was calculated for this system.

**Table 5.** The data of optical density measurements and calculation of stability constant of the 1:1 [CuSal]$^0$ complex in isomolar solutions of the CuCl$_2$—NaHSal (pH 3.5; $V_{cum}$ = 6 mL; $\lambda$ = 750 nm; $D_\infty$ = 0.556; $C^0$ (CuCl$_2$) = $C^0$ (NaHSal) = 2 · 10$^{-2}$ mole/L; log$B_1$ = 13.7; log$B_2$ = 16.53; $K_{h1}$ (Cu$^{2+}$) = 3.1 · 10$^{-8}$; $f$ = 1.92 · 10$^{10}$; $\omega$ = 1.007).

| No. | $V_M$, mL | $V_L$, mL | $D$ | $D_M$ | $\Delta D$ | $C_M$, mole/L | $C_L$, mole/L | $\alpha_C$ | $\beta_1 \cdot 10^{11}$ | log$\beta_1$ |
|---|---|---|---|---|---|---|---|---|---|---|
| 1 | 6.00 | 0.00 | 0.213 | 0.213 | 0.000 | 2.00 · 10$^{-2}$ | 0.00 | – | – | – |
| 2 | 4.00 | 2.00 | 0.175 | 0.142 | 0.033 | 1.30 · 10$^{-2}$ | 6.70 · 10$^{-3}$ | 0.0797 | 2.57 | 11.41 |
| 3 | 3.50 | 2.50 | 0.164 | 0.124 | 0.040 | 1.20 · 10$^{-2}$ | 8.30 · 10$^{-3}$ | 0.0930 | 2.68 | 11.43 |
| 4 | 3.25 | 2.75 | 0.155 | 0.115 | 0.040 | 1.08 · 10$^{-2}$ | 9.16 · 10$^{-3}$ | 0.0907 | 2.28 | 11.36 |
| 5 | 2.75 | 3.25 | 0.141 | 0.098 | 0.043 | 9.16 · 10$^{-3}$ | 1.18 · 10$^{-2}$ | 0.0940 | 1.83 | 11.26 |
| 6 | 2.50 | 3.50 | 0.130 | 0.089 | 0.041 | 8.30 · 10$^{-3}$ | 1.20 · 10$^{-2}$ | 0.0878 | 1.65 | 11.22 |
| 7 | 2.25 | 3.75 | 0.121 | 0.080 | 0.041 | 7.50 · 10$^{-3}$ | 1.25 · 10$^{-2}$ | 0.0861 | 1.55 | 11.19 |
| 8 | 2.00 | 4.00 | 0.112 | 0.071 | 0.042 | 6.70 · 10$^{-3}$ | 1.30 · 10$^{-2}$ | 0.0866 | 1.49 | 11.17 |

log$\beta_1$ = 11.29 ± 0.10.

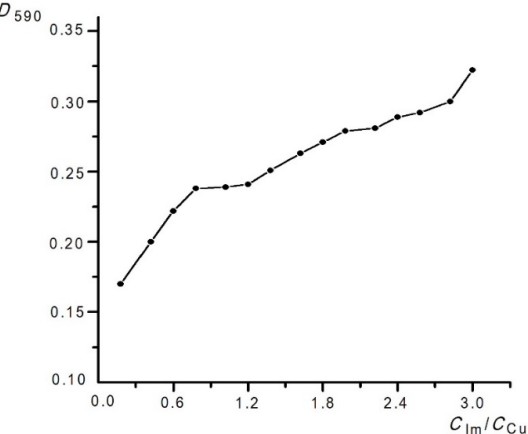

**Figure 2.** Saturation curve of system CuCl$_2$–H$_2$Tar–*x*HIm, $\lambda_{ef}$ = 590 nm ($C^0_{Cu}$ = $C^0_{Tar}$ = $C^0_{HIm}$ = 0.075 mole/L; $C_{Cu}$ = $C_{Tar}$ = 0.0125 mole/L; $I$ = 0.3; pH 6,8; $V_{cum}$ = 6 mL).

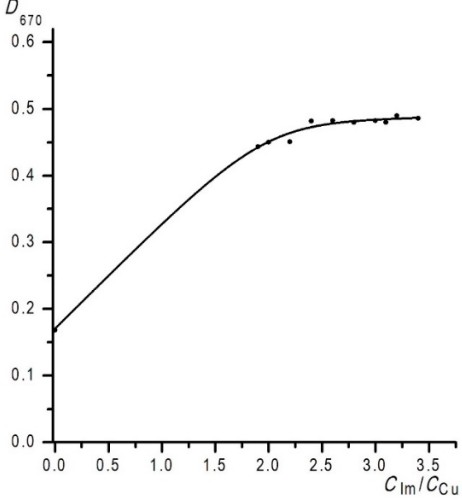

**Figure 3.** Saturation curve of CuCl$_2$–NaHSal–HIm (pH 6.8; $C$ (CuCl$_2$) = $C$ (NaHSal) = 1 · 10$^{-2}$ mole/L; $C$ (HIm) = 0.00÷0.03 mole/L; $V_{cum}$ = 6 mL).

**Table 6.** Calculation results of the stability constant $\lg\beta_{111}$ in the $[CuHImTar]^0$ complex according to data of the saturation curve of $CuCl_2$–$H_2Tar$–$x$HIm ($\lambda_{ef}$ = 670 nm, pH 6.8; $f_{Im}$ = 8.74; $K_{h1}$ ($Cu^{2+}$) = $3.1 \cdot 10^{-8}$; $\log\beta_{CuTart}$ = 3.03; $I$ = 0.3; $V_{cum}$ = 6 mL; $C^0_{HIm}$ = 0.075 mole/L; $C_{Cu}$ = $C_{Tar}$ = 0.0125 mole/L; $\varepsilon_0$ = $D_{CuTar}/C_{Cu}$ = 0.042/0.0125 = 3.36; $D_\infty$ = 0.480; $\varepsilon_\infty$ = 38.4; $C_C$ = $\alpha_\infty C_{Cu}$ ($C_L$)).

| No. | $D_{670}$ | $\varepsilon_i$ | $\alpha_{\infty i}$ | $V_{HIm}$, mL | $C_{HIm}$, mole/L | $C_C$, mole/L | $\log\beta_{111}$ |
|-----|-----------|-----------------|---------------------|---------------|-------------------|---------------|-------------------|
| 1 | 0.369 | 29.52 | 0.747 | 0.2 | $2.500 \cdot 10^{-3}$ | $1.868 \cdot 10^{-3}$ | 7.69 |
| 2 | 0.381 | 30.48 | 0.774 | 0.4 | $5.000 \cdot 10^{-3}$ | $3.870 \cdot 10^{-3}$ | 7.50 |
| 3 | 0.389 | 31.12 | 0.792 | 0.6 | $7.500 \cdot 10^{-3}$ | $5.942 \cdot 10^{-3}$ | 7.41 |
| 4 | 0.445 | 35.6 | 0.920 | 0.8 | $1.000 \cdot 10^{-2}$ | $9.200 \cdot 10^{-3}$ | 7.18 |
| 5 | 0.450 | 36.00 | 0.932 | 1.2 | $1.500 \cdot 10^{-2}$ | $1.164 \cdot 10^{-2}$ | 7.63 |
| 6 | 0.460 | 36.80 | 0.954 | 1.4 | $1.750 \cdot 10^{-2}$ | $1.193 \cdot 10^{-2}$ | 7.59 |
| 7 | 0.462 | 36.96 | 0.959 | 1.6 | $2.000 \cdot 10^{-2}$ | $1.199 \cdot 10^{-2}$ | 7.49 |

$\log\beta_{111}$ = 7.49 ± 0.16.

In Tables 6 and 7, the data for calculating the stability constants $\beta_{111}$ of the [CuHImTar] complex and $\beta_{121}$ of the $[Cu(HIm)_2Sal]$ complex, respectively, are presented. The stability constants of MLC were calculated according to the procedure described in [26]. For equilibrium, for example, with the participation of the two complexes $Cu(HIm)_2L$ and CuL ($L^{2-}$ − $Sal^{2-}$) absorbing with the same wavelength,

$$CuL + 2HIm \overset{K}{\leftrightarrow} Cu(HIm)_2L$$

(charges are omitted for convenience), the equilibrium constant $K$ connected with the stability constants $\beta_1$ of the monoligand CuL complex and $\beta_{121}$ of the mixed-ligand complex $Cu(HIm)_2L$ by the ratio of $\beta_{121} = K \cdot \beta_1$. When using photometric data (Table 7) for each point of the saturation curve, we have

$$\beta_{121} = \beta_1(\alpha_\infty f^2_{HIm})/((1 - \alpha_\infty) \times (C_{HIm} - 2\alpha_\infty C_{Cu})^2),$$

where $\alpha_\infty$ is the maximal yield of the $Cu(Him)_2L$ complex; $\alpha_\infty = (\varepsilon_i - \varepsilon_0)/(\varepsilon_\infty - \varepsilon_0)$; $\varepsilon_\infty = D_\infty/C_{Cu}$, $\varepsilon_i = D_i/C_{Cu}$; $\varepsilon$ is the molar absorption factor of the corresponding particles: CuL ($\varepsilon_0$), $Cu(Him)_2L$ ($\varepsilon_\infty$), and CuL + $Cu(HIm)_2L$ ($\varepsilon_i$); and $f_{HIm} = 1 + B_1[H^+]$ ($\log B_1$ = 7.69, according to the data in [27]). The stability constant of the salicylate complex [CuSal] ($\lg\beta_1$ = 11.27) was borrowed from this work (the solubility method). The stability constant of the tartrate complex of copper(II) [CuTar] ($\log\beta_1$ = 3.03) was taken from [9].

**Table 7.** Calculation results of stability constant $\lg\beta_{121}$ in the $[Cu(HIm)_2Sal]^0$ complex according to data of the saturation curve of $CuCl_2$–$NaHSal$–$x$HIm (pH = 6.8; $I$ = 0.3; $C_{Cu}$ = $C_{Sal}$ = $1 \cdot 10^{-2}$ mole/L; $V_{cum}$ = 6 mL; $C^0$ (HIm) = $6 \cdot 10^{-2}$ mole/L; $\varepsilon_0$ = 16.8; $\varepsilon_\infty$ = 49.0; $f_{HIm}$ = 8.77; $\beta_{CuSal}$ = $1.86 \cdot 10^{11}$; $D_\infty$ = 0.490; $C_C$ = $\alpha_\infty C_{Cu}$; $K_{h1}$ ($Cu^{2+}$) = $3.1 \cdot 10^{-8}$).

| No. | $D_{670}$ | $\varepsilon_i$ | $\alpha_{\infty i}$ | $V_{HIm}$, mL | $C_{HIm}$, mole/L | $C_C$, mole/L | $\log\beta_{121}$ |
|-----|-----------|-----------------|---------------------|---------------|-------------------|---------------|-------------------|
| 1 | 0.450 | 45.0 | 0.876 | 2.0 | $2.00 \cdot 10^{-2}$ | $8.76 \cdot 10^{-3}$ | 19.22 |
| 2 | 0.451 | 45.1 | 0.879 | 2.1 | $2.10 \cdot 10^{-2}$ | $8.79 \cdot 10^{-3}$ | 18.96 |
| 3 | 0.480 | 48.0 | 0.969 | 2.8 | $2.80 \cdot 10^{-2}$ | $9.70 \cdot 10^{-3}$ | 18.79 |
| 4 | 0.486 | 48.6 | 0.986 | 3.4 | $3.40 \cdot 10^{-2}$ | $9.86 \cdot 10^{-3}$ | 18.71 |
| 5 | 0.483 | 48.3 | 0.978 | 2.6 | $2.60 \cdot 10^{-2}$ | $9.79 \cdot 10^{-3}$ | 19.10 |
| 6 | 0.483 | 48.3 | 0.978 | 3.0 | $3.00 \cdot 10^{-2}$ | $9.79 \cdot 10^{-3}$ | 18.78 |

$\log\beta_{121}$ = 18.94 ± 0.23.

Electronic absorption spectra in the visible region (Figures 4 and 5) were recorded for the aqueous systems $Cu^{2+}$–$H_2O$, $Cu^{2+}$–$L^{2-}$, $Cu^{2+}$–HIm, and $Cu^{2+}$–$L^{2-}$–(1–2)HIm ($L^{2-}$ is tartrate and salicylate anions).

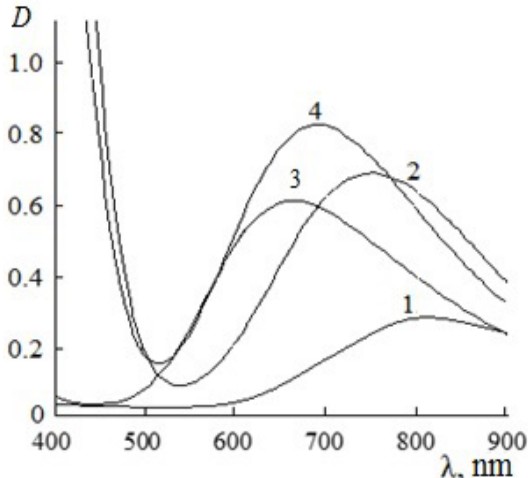

**Figure 4.** Electronic absorption spectra of systems: 1—$CuCl_2$−$H_2O$ ($C_{Cu}$ = 0.025 mole/L; pH 5.50); 2 —$CuCl_2$−$Na_2$Tar ($C_{Cu}$ = $C_{Tar}$ = 0.025 mole/L; pH 6.00); 3—$CuCl_2$−HIm ($C_{Cu}$ = $C_{HIm}$ = 0.025 mole/L; pH 6.50); 4—$CuCl_2$–$Na_2$Tar–HIm ($C_{Cu}$ = $C_{Tar}$ = $C_{Him}$ = 0.025 mole/L; pH 6.50).

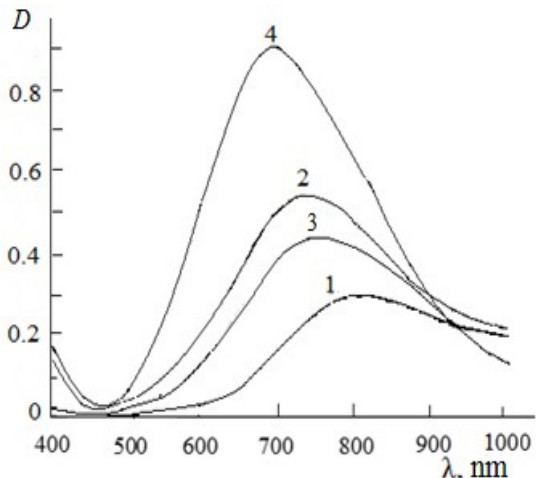

**Figure 5.** Electronic absorption spectra of systems: 1—$CuCl_2$–$H_2O$ ($C_{Cu}$ = 0.02 mole/L; pH 4.00); 2—$CuCl_2$–$Na_2$Sal ($C_{Cu}$ = $C_{Sal}$ = 0.02 mole/L; pH 4.95); 3—$CuCl_2$–HIm ($C_{Cu}$ = 0.02 mole/L; $C_{Him}$ = 0.04 mole/L; pH 5.95); 4—$CuCl_2$–$Na_2$Sal–HIm ($C_{Cu}$ = $C_{Sal}$ = 0.02 mole/L; $C_{HIm}$ = 0.04 mole/L; pH 5.33).

## 3. Discussion

Synthesized original salts of copper(II) CuTar · $2H_2O$ and CuSal · $H_2O$ represent fine crystalline substances slightly soluble in water (determined by the authors for these salts under ionic strengths of 0.3 log$K_S$ = −7.44 and log$K_S$ = −13.61, respectively). Mixed-ligand salts, containing the oxycarboxylic acid anion and a neutral molecule of azole as ligands, were synthesized using the slightly soluble tartrate and salicylate of copper(II) and azoles, with the pH of the aqueous solution equal to 6.5–8.5. Figure 6 shows a distribution diagram of the neutral imidazole HIm molecule (the mentioned hydrogen atom belongs to the pyrrole nitrogen atom $N_{(1)}$, $pK_a$ = 14.5) and the protonated $H_2Im^+$ form [28]. The neutral imidazole HIm molecule (curve 1) has a wide area of dominance in the pH range of 5.5–13.0. Therefore, within these pH limits, the pyridine nitrogen $N_{(3)}$ atom of the neutral HIm

molecule is assumed to participate in the reactions of the formation of metal complexes, since it contains an unshared electron pair on the $sp^2$-hybrid orbital of the nitrogen atom. This is confirmed by the study of the IR spectra of MLS. The neutral imidazole molecule coordination by a copper ion is also possible during a competitive reaction:

$$H_2lm^+ + M^{n+} \leftrightarrow MHlm^{n+} + H^+$$

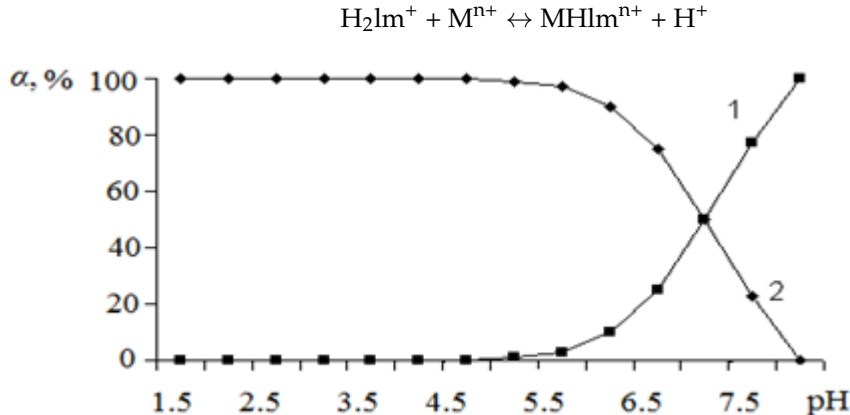

**Figure 6.** Diagram of the yield of imidazole particles in the aqueous solution: 1—HIm, 2—$H_2Im^+$.

The synthesis of mixed-ligand salts was performed at different CuL:HIm((metIm) ratios. Table 1 shows that, under the stated synthesis conditions, tartrate and copper(II) salicylate coordinate three imidazole molecules and two 2-methylimidazole molecules.

To determine the functional groups participating in complex formation with copper(II) ions, the infrared spectra of the ligands and mono- and bi-ligand salts were used. The IR spectra of tartaric and salicylic acids are characterized by narrow intensive absorption bands of the carbonyl group C=O at 1711.3 and 1654.8 cm$^{-1}$, respectively. The spectra of the mono- and bi-ligand salts do not have such bands. However, they contain intense bands of asymmetric and symmetric valence vibrations of the ionized carboxyl group COO$^-$ involved in coordination with the metal cation. They were 1610.5 and 1322.5 cm$^{-1}$ for CuTar · 2H$_2$O; 1588.8 and 1327.2 cm$^{-1}$ for Cu(HIm)$_3$Tar · 2H$_2$O; 1622.5 and 1386.4 cm$^{-1}$ for CuSal · H$_2$O; 1628.6 and 1384.4 cm$^{-1}$ for Cu(HIm)$_3$Sal; and 1600.6 and 1371.2 cm$^{-1}$ for Cu(metIm)$_2$Sal. This confirmed the connection in the salts of the copper ion with the ionized carboxyl groups of tartaric and salicylic acids. The absorption band, corresponding to the deformation vibrations of the salicylic acid oxygroup at 1480.9 cm$^{-1}$ in the spectrum of copper(II) salicylate, shifted to 1448.4 cm$^{-1}$, and in the biligand salts, it shifted to 1458 cm$^{-1}$ (imidazolesalicylate) and 1445.2 cm$^{-1}$ (methylimidazolesalicylate). This enabled the conclusions that the salicylic acid oxygroup participates in connection with the copper(II) cation in mono- and bi-ligand salts and that salicylic acid in synthesized salts behaves as bicarboxylic acid.

A shift of the intensive bands of the deformation vibrations and out-of-plane vibrations of the imidazole ring (657.9, 1053.9 cm$^{-1}$) in the IR spectrum of the biligand salts (654 and 1064 cm$^{-1}$ in imidazoletartrate; 650 and 1069 cm$^{-1}$ in imidazolesalicylate) was a confirmation of the imidazole coordination in the internal sphere of the complex. The absence of the absorption bands of the valence vibrations of the bonds C=C and C=N of imidazole (1829.0 and 1770.6 cm$^{-1}$) in the IR spectra of the biligand salts indicated a connection between a Cu$^{2+}$ ion and the N$_{(3)}$ atom of imidazole.

Table 8 shows that the stability constant value of the monoligand complex [CuSal] (log$\beta_1$ = 11.27; log$\beta_1$ = 11.29), which we determined using two methods, consistently conformed satisfactorily to the value offered by the literature. The agreement of the log$\beta$1 values for the [CuTar] complex was worse, owing to the solubility method having a lower accuracy than other methods.

**Table 8.** Results of determining stability constants of mono- and bi-ligand complexes of copper(II) with anions of tartaric and salicylic acids and imidazole in the solution.

| Composition of Complex | Present Work | | Source, Value Log$\beta_1$ |
|---|---|---|---|
| | Determination Method | Log$\beta_1$, Log$\beta_{1i1}$ | |
| [CuTar] | Solubility | 3.74 ($s^2 = 2.68 \cdot 10^{-3}$) | [9], 3.03; [10], 3.1; [8], 2.7 |
| [CuSal] | Solubility; isomolar series | 11.27($s^2 = 1.77 \cdot 10^{-3}$); 11.29 $\pm$ 0.10 | [10], 10.6 |
| [CuHImTar] | Saturation curve | 7.49 $\pm$ 0.16 | – |
| [Cu(HIm)$_2$Sal] | Saturation curve | 18.94 $\pm$ 0.23 | – |

The saturation curves of the ternary systems $Cu^{2+}$–Him–$L^{2-}$ (Figures 2 and 3) showed the formation of complexes of compositions 1:1:1 (CuHImTar) and 1:2:1 (Cu(Him)$_2$Sal) in the solution. The obtained stability constant of MLC of copper(II) with tartaric acid and imidazole (Table 8) was used to construct a diagram of the yield of particles from pH in the studied system at a molar ratio of components of 1:1:1 ($C_{Cu} = C_{Tar} = C_{HIm}$ = 0.0125 mole/L; pH interval of 0–9). The calculation of the equilibrium composition of the solution and the construction of the diagram (Figure 7) were carried out using the HySS2009 program [29] taking into account Equilibria (1)–(10) and the corresponding equilibrium constants:

| Equilibrium | Equilibrium Constant Logarithm |
|---|---|
| $Tar^{2-} + H^+ \leftrightarrow HTar^-$ | (1), lg$B_{1T}$ = 3.95 |
| $Tar^{2-} + 2H^+ \leftrightarrow H_2Tar$ | (2), lg$B_{2T}$ = 6.76 |
| $HIm + H^+ \leftrightarrow H_2Im^+$ | (3), lg$B_{1I}$ = 7.69 |
| $Cu^{2+} + Tar^{2-} \leftrightarrow CuTar$ | (4), lg$\beta_{1T}$ = 3.10 [10] |
| $Cu^{2+} + 2Tar^{2-} \leftrightarrow CuTar_2{}^{2-}$ | (5), lg$\beta_{2T}$ = 5.11 [30] |
| $Cu^{2+} + HIm \leftrightarrow CuHIm^{2+}$ | (6), lg$\beta_{1I}$ = 4.33 [10] |
| $Cu^{2+} + 2HIm \leftrightarrow Cu(HIm)_2{}^{2+}$ | (7), lg$\beta_{2I}$ = 7.57 [18] |
| $Cu^{2+} + Tar^{2-} + HIm \leftrightarrow CuTarHIm$ | (8), lg$\beta_{111}$ = 7.49 |
| $Cu^{2+} + H_2O \leftrightarrow CuOH^+ + H^+$ | (9), lg$K_{h1}$ = $-7.53$ |
| $H_2O \leftrightarrow H^+ + OH^-$ | (10), lg$K_W$ = $-13.8$ |

Figure 7 shows that the mixed-ligand CuHImTar particles turn out to be the dominant forms in a wide pH range (curve 4). The compatibility in the inner sphere of the MLC of the copper(II) of the ligands, containing donor nitrogen and oxygen atoms, provides a high yield of MLC (more than 60%).

The results of [18,19] and the present research (Table 8) show that the attachment of one molecule of imidazole to the copper ion increases the stability constant logarithm of the complex ion by approximately 3.5–4 units. The authors in [31] estimate the compatibility of different ligands (L, A) in the inner sphere of the complex with a 1:1:1 composition by using the value of the $k_S$ co-proportioning constant, which is related to the common stability constants of the complexes by the ratio of lg$k_S$ = lg$\beta$(MLA) $- \frac{1}{2}$(lg$\beta$(ML$_2$) + lg$\beta$(MA$_2$)). When the ligands are compatible, $k_S > 1$, and the MLC stability constant is greater than the arithmetic mean among the stability constants of the monoligand complexes. The ligands can be shown to be compatible in the CuHImTar complex since lg$k_S$ = 7.49 $-$ (0.5 $\cdot$ 7.57 + 0.5 $\cdot$ 5.11) = 1.15 and $k_S > 1$ (lg$\beta$(Cu(HIm)$_2$) = 7.57 [18], lg$\beta$(CuTar$_2$) = 5.11 [30]), and lg$\beta$(CuHImTar) $> \frac{1}{2}$ (lg$\beta$(Cu(HIm)$_2$) + lg$\beta$(CuTar$_2$)), 7.49 > 6.34.

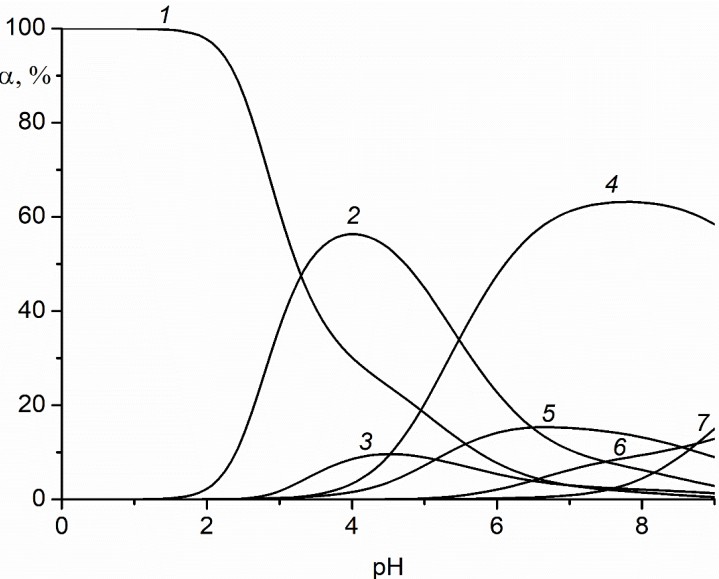

**Figure 7.** Diagram of particle yield from pH in the system $Cu^{2+}$–$H_2Tar$–HIm: 1—$[Cu^{2+}]$, 2—[CuTar], 3—$[CuTar_2]^{2-}$, 4—[CuTarHIm], 5—$[CuHIm]^{2+}$, 6—$[Cu(HIm)_2]^{2+}$, 7—$[CuOH]^+$ ($C_{Cu} = C_{Tar} = C_{HIm}$ = 0.0125 mole/L).

The electronic absorption spectra of the systems (Figures 4 and 5) confirm the formation of MLC in the solution. The large value of optical density $D$ in the system with the two ligands $CuCl_2$–$H_2L$–HIm (Figures 4 and 5, curve 4) as compared to the systems $CuCl_2$–HIm and $CuCl_2$–L is evidence of the formation of new complexes. This was also confirmed by the shifting of the absorption maximums of the systems with complexes, compared to the hydrated copper ion, to the short-wave region. The substitution of water molecules in the coordination sphere of the $Cu^{2+}$ ion by more firmly bound ligands (the best donors of electronic pairs) increases the difference in the energies of the split *d*-sublevels of the complex former. Its *d-d*-band of absorption shifts to the side of shorter wavelengths (the hypsochromic effect).

The formation of mixed-ligand solid salts and complexes in the solution is related to the affinity of the $Cu^{2+}$ ($d^9$) ion to both the donor nitrogen atoms and the oxygen atoms of ligands — imidazole and the anions of oxycarboxylic acids. The obtained biligand salts of oxycarboxylic acids, imidazole, and 2-methylimidazole hold promise when used as substances containing a metal microelement and bioactive ligands.

## 4. Conclusions

A method for the synthesis of new mixed-ligand copper(II) compounds of the composition $Cu(HIm)_x(metIm)_xTar(Sal)$ from low-soluble copper(II) oxycarboxylates and azoles was developed. The composition of these compounds was confirmed using chemical, thermal, and thermogravimetric methods. It is planned to study the various biological properties of the obtained compounds in comparison with those of the initial components—copper(II) salicylate (antiseptic) and azoles (antimycotics).

An analysis of the electronic absorption spectra of the systems confirmed the formation of MLC in an aqueous solution, and an analysis of the IR spectra of the initial ligands and selected mixed-ligand copper(II) compounds allowed us to conclude the structure of the latter: the complexation with copper(II) ions involves the nitrogen atom $N_{(3)}$ of imidazole, the oxygen atoms of the carboxyl groups of tartaric acid, and the oxygen atoms of the carboxyl and hydroxyl groups of salicylic acid. The formation of individual mixed-ligand compounds and complexes in a solution with azole molecules and oxycarboxylic acid anions is due to the affinity of the $Cu^{2+}$ ion ($d^9$) for donor nitrogen and oxygen atoms.

The solubility constants of copper(II) tartrate and salicylate and the stability constants of the mono- and bi-ligand complexes ([CuTar(Sal)], [CuHImTar], and [Cu(HIm)$_2$Sal])

determined in this work complete the bank of thermodynamic values for copper(II) salts and complexes.

**Author Contributions:** N.S.: conceptualization, chemical synthesis, methodology, investigation, formal analysis, data curation, validation. E.T.: investigation, formal analysis. E.P.: investigation, formal analysis. I.K.: methodology, formal analysis, validation, data curation, writing the manuscript, reviewing and editing. All authors have read and agreed to the published version of the manuscript.

**Funding:** The reported study was funded by Tomsk State University Development Program (Priority-2030).

**Data Availability Statement:** The authors confirm that all of the data in the article have not been published elsewhere and are available in the article itself.

**Conflicts of Interest:** The authors declare no conflict of interest. The founding sponsors had no role in the design of the study; in the collection, analyses, or interpretation of the data; in the writing of the manuscript; or in the decision to publish the results.

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
