# Peer review of "Interaction of Copper(II) Ions with Certain Oxyacids and Azoles"

_inorganics, doi:10.3390/inorganics11060232_

Round 1
Reviewer 1 Report
The manuscript presented by Irina Kurzina and co-workers deals with the preparation of a series of mono-ligand and mixed-ligand Cu(II) based salts using either tartaric acid or salicylic acid as well as imidazole and 2-methylimidasole as representative examples for azole ligands. After the synthesis, the authors studied a range of parameters including the composition in the solid state and in solution, the solubility behavior, stability constants, the pH-dependency etc. While the study appears systematic and robust, there are several open questions and recommendations for improvement.
Please, improve the following points prior to publication in Inorganics:
1.) Potential applications and fields of use are described in the introduction, but unfortunately not demonstrated. The whole study and their impact would greatly benefit, if the authors would show a real application of their salts and results. Maybe some preliminary results at the end would be fine.
2.) The authors applied different techniques (e.g. elemental analysis, IR spectroscopy) to study the structure of the synthesized salts. However, X-ray techniques are completely missing. Therefore, I encourage the authors to also provide some X-ray measurements (e.g. powder XRD or single crystal measurements or crystalline samples).
3.) Line 97: “The reagents used were labeled as c.p. or a.p.”. What does c.p. and a.p. exactly mean in this context?
4.) Table 5: The applied abbreviations and constants should be explained in more detail and provide along with Table 5. Maybe add this information as a footnote to Table 5.
5.) Figure 5: The X-axis looks rather uncommon for UV/vis spectra. Typically, the absorbance is plotted as a function of wavelength. Why are the spectra cut off below 400 nm? What causes the very strong increase at about 450 nm?
6.) Figure 7: The arrows for an equilibrium reaction look very strange. Why is there a box around? Please, improve this.
7.) Representative IR spectra should be presented in the main manuscript.
8.) Some references have DOI numbers and links, but others not. Furthermore, some links seem wrong and not to work. Please, check this carefully and provide all DOI numbers.
Author Response
The manuscript presented by Irina Kurzina and co-workers deals with the preparation of a series of mono-ligand and mixed-ligand Cu(II) based salts using either tartaric acid or salicylic acid as well as imidazole and 2-methylimidasole as representative examples for azole ligands. After the synthesis, the authors studied a range of parameters including the composition in the solid state and in solution, the solubility behavior, stability constants, the pH-dependency etc. While the study appears systematic and robust, there are several open questions and recommendations for improvement.
Please, improve the following points prior to publication in Inorganics:
1.) Potential applications and fields of use are described in the introduction, but unfortunately not demonstrated. The whole study and their impact would greatly benefit, if the authors would show a real application of their salts and results. Maybe some preliminary results at the end would be fine.
Answer 1. To show the real application of the salts and the results require new research and time. It is planned to investigate biological properties (antibacterial, anti-hypoxic, toxicity, cytotoxicity, etc.) of imidazole(HIm), 2-methylimidazole(metIm)-salicylate salts of copper(II) composition Cu(C3H4N2)3C7H4O3, Cu(C4H6N2)2C7H4O3 in comparison with the initial compounds - copper(II) salicylate (antiseptic) and azoles (antifungal).
2.) The authors applied different techniques (e.g. elemental analysis, IR spectroscopy) to study the structure of the synthesized salts. However, X-ray techniques are completely missing. Therefore, I encourage the authors to also provide some X-ray measurements (e.g. powder XRD or single crystal measurements or crystalline samples).
Answer 2. It is difficult to perform X-ray measurements at this time.
3.) Line 97: “The reagents used were labeled as c.p. or a.p.”. What does c.p. and a.p. exactly mean in this context?
Answer 3. The correction in the text (p. 3) is done (highlighted in red-brick).
4.) Table 5: The applied abbreviations and constants should be explained in more detail and provide along with Table 5. Maybe add this information as a footnote to Table 5.
Answer 4. Notations and constants are explained in the text of the article (p. 3; 6 after the reference to Table 5); designations D, DM, ∆D (see Fig. 1), VM, VL, CM, SL are common in analytical chemistry, chemistry of complex compounds, etc.
5.) Figure 5: The X-axis looks rather uncommon for UV/vis spectra. Typically, the absorbance is plotted as a function of wavelength. Why are the spectra cut off below 400 nm? What causes the very strong increase at about 450 nm?
Answer 5. Fig. 5 shows the dependence of the optical density of the solutions on the wavelength. Since the Cu2+(d9) complexes under study are stained, the working region of the spectrum (500-900) nm (the region of d-d-transitions in copper(II) complexes) is highlighted in the figure.
6.) Figure 7: The arrows for an equilibrium reaction look very strange. Why is there a box around? Please, improve this.
Answer 6. The arrows for equilibrium (1-10) are plotted (probably an error occurred due to technical reasons during text transfer).
7.) Representative IR spectra should be presented in the main manuscript.
Answer 7. The original IR spectra cannot be presented. The results of the analysis of the spectra are available on p. 10 of the manuscript. 10 of the manuscript.
8.) Some references have DOI numbers and links, but others not. Furthermore, some links seem wrong and not to work. Please, check this carefully and provide all DOI numbers.
Answer 8. DOIs are given for all articles in which they are available.

Reviewer 2 Report
The manuscript by Kurzina et. al. present the synthesis and characetrization of several novel copper complexes with tartrate and imidazoles. The material can be published in Inorganics, however, several serious issues should be addressed first.
1) a clear synthetic part should be added with procedures and characterization data for all new compounds.
2) The language needs to be corrected.
3) The conclusion section needs to be improved. It more descriptive than conclusive. The authors should give something more than constation of experimental facts.
If this critical issues are resolved, than the manuscript can be accepted.
Author Response
Responses to reviewer 2
The manuscript by Kurzina et. al. present the synthesis and characetrization of several novel copper complexes with tartrate and imidazoles. The material can be published in Inorganics, however, several serious issues should be addressed first.
1) a clear synthetic part should be added with procedures and characterization data for all new compounds.
Answer 1. We believe that the syntheses of new salts of the composition Cu(HIm)x(metIm)xTar(Sal) are described (pp. 4-5) clearly enough that these syntheses can be reproduced.
2) The language needs to be corrected.
Answer 2. The text of the article has been checked and corrected for grammatical errors and typos.
3) The conclusion section needs to be improved. It more descriptive than conclusive. The authors should give something more than constation of experimental facts.
Answer 3. The conclusion of the article is corrected, added to the text.
- Conclusion
A method for the synthesis of new mixed copper(II) ligand compounds of the composition Cu(HIm)x(metIm)xTar(Sal) from low-soluble copper(II) oxycarboxylates and azoles has been developed. The composition of these compounds was confirmed by chemical, thermal, and thermogravimetric methods. It is planned to study various biological properties of obtained compounds in comparison with the initial components - copper(II) salicylate (antiseptic) and azoles (antimycotic),
Analysis of electronic absorption spectra of the systems confirmed the formation of MLC in aqueous solution, and analysis of IR spectra of the initial ligands and selected mixed-ligand copper(II) compounds allowed us to conclude on the structure of the latter: the complexation with copper(II) ion involves nitrogen atom N(3) of imidazole, oxygen atoms of carboxyl groups of tartaric acid, oxygen atoms of carboxyl and hydroxyl groups of salicylic acid. The formation of individual mixed ligand compounds and complexes in solution with azole molecules and oxycarboxylic acid anions is due to the affinity of the Cu2+ ion (d9) for donor nitrogen and oxygen atoms.
The solubility constants of copper(II) tartrate and salicylate, stability constants of mono- and biligand complexes ([CuTar(Sal)], [CuHImTar], [Cu(HIm)2Sal]) determined in this work complete the bank of thermodynamic values for copper(II) salts and complexes.

Round 2
Reviewer 2 Report
The manuscript can be accepted in the present form.